

# Structural and functional brain changes in people with knee osteoarthritis: a scoping review

Joaquín Salazar-Méndez[1], Iván Cuyul-Vásquez[2,3], Nelson Viscay-Sanhueza[4], Juan Morales-Verdugo[5], Guillermo Mendez-Rebolledo[1], Felipe Ponce-Fuentes[6] and Enrique Lluch-Girbés[7]

[1] Laboratorio de Investigación Somatosensorial y Motora, Escuela de Kinesiología, Facultad de Salud, Universidad Santo Tomás, Talca, Chile
[2] Departamento de Procesos Terapéuticos, Facultad de Ciencias de la Salud, Universidad Católica de Temuco, Temuco, Chile
[3] Facultad de las Ciencias de la Salud, Universidad Autónoma de Chile, Temuco, Chile
[4] Unidad de medicina física y rehabilitación, Hospital Dr. Gustavo Fricke, Viña del Mar, Chile
[5] Departamento de Ciencias Preclínicas, Facultad de Medicina, Universidad Católica del Maule, Talca, Chile
[6] Facultad de Medicina y Ciencias de la Salud, Escuela de Kinesiología, Universidad Mayor, Temuco, Chile
[7] Department of Physiotherapy, Faculty of Physiotherapy, University of Valencia, Valencia, Spain

## ABSTRACT

**Background**. Knee osteoarthritis is a highly prevalent disease worldwide that leads to functional disability and chronic pain. It has been shown that not only changes are generated at the joint level in these individuals, but also neuroplastic changes are produced in different brain areas, especially in those areas related to pain perception, therefore, the objective of this research was to identify and compare the structural and functional brain changes in knee OA *versus* healthy subjects.
**Methodology**. Searches in MEDLINE (PubMed), EMBASE, WOS, CINAHL, SCOPUS, Health Source, and Epistemonikos databases were conducted to explore the available evidence on the structural and functional brain changes occurring in people with knee OA. Data were recorded on study characteristics, participant characteristics, and brain assessment techniques. The methodological quality of the studies was analysed with Newcastle Ottawa Scale.
**Results**. Sixteen studies met the inclusion criteria. A decrease volume of the gray matter in the insular region, parietal lobe, cingulate cortex, hippocampus, visual cortex, temporal lobe, prefrontal cortex, and basal ganglia was found in people with knee OA. However, the opposite occurred in the frontal lobe, nucleus accumbens, amygdala region and somatosensory cortex, where an increase in the gray matter volume was evidenced. Moreover, a decreased connectivity to the frontal lobe from the insula, cingulate cortex, parietal, and temporal areas, and an increase in connectivity from the insula to the prefrontal cortex, subcallosal area, and temporal lobe was shown.
**Conclusion**. All these findings are suggestive of neuroplastic changes affecting the pain matrix in people with knee OA.

Corresponding author
Guillermo Mendez-Rebolledo, guillermomendezre@santotomas.cl

## INTRODUCTION

Knee osteoarthritis (OA) is the most common joint condition (*Vos et al., 2012*), due to wear of the articular cartilage affecting all three compartments of the knee (medial, lateral, and patellofemoral joint) (*Roos & Arden, 2016*; *Lawson et al., 2022*), and is considered a progressive multifactorial disease (*Hunter & Bierma-Zeinstra, 2019*). The global prevalence of knee OA reaches 16.0% in people aged 15 years or older and 22.9% in people older than 40 years with incidence rates over 20 years of 203 per 10,000 people annually (*Cui et al., 2020*). Pain, the main symptom of OA of the knee (*Parks et al., 2011*), is associated with dependence on healthcare systems (*Peat, McCarney & Croft, 2001*), a decrease in quality of life (*Salaffi et al., 2005*), a deterioration in physical function, and an increased risk of disability (*Jinks, Jordan & Croft, 2007*).

Although it is true that knee OA is classified as a peripheral joint disease, it has been shown in these patients that the perception of pain intensity does not necessarily correlate with the joint damage they present (*Kurien et al., 2018*; *Simis et al., 2021*; *Iuamoto et al., 2022*) and even persists in those who undergo surgery (*Baker et al., 2007*; *Kurien et al., 2022*). This is because pain processing is subjective and is mediated by both peripheral and central mechanisms (*Baliki et al., 2014*; *Fingleton et al., 2015*; *Fu, Robbins & McDougall, 2018*). In this sense, neuroplastic changes have been identified in the central nervous system at the spinal cord, brainstem, and brain level (*Apkarian, Hashmi & Baliki, 2011*), related to prolonged duration of pain (*Pelletier, Higgins & Bourbonnais, 2015*; *Alshuft et al., 2016*; *Skou et al., 2016*).

There is a growing body of evidence that has paid special attention to changes at the brain level, pointing to the presence of structural plasticity and an important functional brain reorganization in chronic musculoskeletal conditions assessed mainly by magnetic resonance imaging (MRI) and electroencephalography (EEG) (*Apkarian, Baliki & Geha, 2009*; *Kuner & Flor, 2016*; *Segning et al., 2022*). Structural plasticity gives us information on volumetric changes, mainly area and thickness (*Kregel et al., 2015*), while, within the functional changes, the functional activity allows us to know the behavior in a specific area (*Herzberg & Gunnar, 2020*) and functional connectivity (FC) allows us to estimate patterns of interregional neuronal interactions (*Lurie et al., 2020*).

In this sense, it has been indicated that both the structure and the function of the brain are affected in patients with knee OA in areas involved in sensory discrimination, as well as affective and cognitive-evaluative areas (*Soni et al., 2019*). For example, gray matter abnormalities have been found in the lateral prefrontal cortex, the parietal lobe, the anterior cingulate cortex, the insula, and the limbic cortex in patients with KOA (*Parks et al., 2011*; *Howard et al., 2012*; *Hiramatsu et al., 2014*). It has also been found that, in other chronic musculoskeletal conditions, there are global and specific alterations in the gray matter, mainly in the prefrontal regions, anterior insula, and cingulate cortex, basal ganglia, thalamus, periaqueductal gray matter, pre and postcentral gyri and inferior parietal lobe (*Cauda et al., 2014*). However, there are no reviews available that clarify the structural and functional brain changes by comparing patients with knee OA with healthy subjects. It is essential to identify the affected areas, the specific changes that occur, and their direction

in order to comprehend the underlying pathophysiology and potentially establish a brain biomarker for chronic pain in patients with knee OA (*Tracey, Woolf & Andrews, 2019*; *Davis et al., 2020*).

Considering the aforementioned factors, including the variability of techniques used in the analyses and the involvement of different brain areas, it becomes essential to conduct a synthesis of the available information in the literature. To achieve this, a scoping review was deemed appropriate and selected as the suitable method. This approach will provide a comprehensive overview of the research landscape and help identify any existing gaps in this particular area of study (*Munn et al., 2018*). The objective of this review was to examine the available evidence on the structural and functional brain changes occurring in people with knee OA in comparison with healthy controls. This information is valuable across a wide range of knowledge in the field of pain neuroscience, benefiting researchers interested in brain neuroplastic changes as well as healthcare providers involved in pain management.

## METHODS

### Design

The PRISMA extension for scoping reviews (*Tricco et al., 2018*) was followed in this study. The framework described by *Arksey & O'Malley (2005)* was utilized. The protocol was registered in the OSF Registries (https://osf.io/eqth8/).

### Search strategy

A systematic literature research was conducted in MEDLINE (via PubMed), EMBASE, Web of Science, Cumulative Index to Nursing and Allied Health Literature (CINAHL), SCOPUS, Health Source, and Epistemonikos databases from inception to July 2022. Detailed search strategy can be found in Supplementary Material 1. In addition, a manual search of the references of the selected articles was performed to identify possible relevant studies.

### Screening and study eligibility criteria

Two researchers (NV-S and JM-V) independently used the systematic review manager Rayyan (http://www.rayyan.ai) (*Ouzzani et al., 2016*) to select potential studies based on title and abstract. A third reviewer (GM-R) resolved any discrepancies. The same process was performed for full-text screening performed in those studies where the title and abstract did not provide enough information.

The studies were included if they presented the following inclusion criteria: adults ≥18 years of age with a diagnosis of knee OA based on the American College of Rheumatology classification (*Wu et al., 2005*), the Chinese Guidelines for Diagnosis and Treatment of Osteoarthritis (*Zhang et al., 2020*), or medical criteria (without specifying the use of guidelines or classification), and graded in severity by using the Kellgren and Lawrence classification (*Kohn, Sassoon & Fernando, 2016*); presenting a transversal design where comparisons were made between people with knee OA and healthy controls, and pre-experimental studies (only the baseline was considered for the comparison); reporting at least one outcome variable regarding structural and/or functional brain
changes as determined by imaging techniques such as magnetic resonance imaging (MRI), electroencephalography (EEG), and positron emission tomography (PET).

Studies were excluded if they included other chronic visceral or cancer pain conditions or were study protocols, conference proceedings, or case report studies.

Only studies written in English or Spanish were considered in this review.

## Data extraction

Data extraction was conducted using a standardized form. The following data were extracted from each article: First author and year of publication, study design, sample characteristics, diagnostic criteria for knee OA, imaging technique employed to assess brain changes and results of the study. Data extraction was performed separately by two reviewers (JS-M and GM-R).

## Quality of evidence assessment

Two investigators independently (JM-V and JS-M) used the Newcastle-Ottawa Scale (NOS) to assess the quality of included studies, which is a validated and easy-to-use 8-item scale in three domains: selection, comparability, and exposure/outcome. Studies receive a score of one point for each item in the selection and exposure/outcome domain, while for the comparability domain there are scores up to two. Studies are scored from 0 to 9, and those studies are scored from 0 to 2 (poor quality), 3 to 5 (fair quality), 6 to 9 (good/high quality) (*Wells et al., 2012*).

# RESULTS

## Study selection

A total of 1,570 studies were retrieved in the databases (Fig. 1). After removing duplicates, and excluding articles based on title and abstract, 24 studies qualified for full-text screening. Of these articles, eight were excluded leaving a total of 16 studies included in the review (Fig. 1). A total sample of 1,119 participants (620 with KOA and 499 healthy controls) was analysed. The number of participants per study is shown in Table 1.

## Quality of evidence assessment

Only six studies presented good/high methodological quality, which represents 37.5% of the research included in this review, while 10 studies were classified as fair quality, which represents 62.5% of the studies. Regarding the evaluation by items, the one that presented the least consideration was the representativeness of the cases (15 studies did not obtain a score). The same method of ascertainment was used for cases and controls (12 studies did not obtain a score) and selection of controls (11 studies did not obtain a score) (see Table 2 for more details).

## Study characteristics

Among the 16 included studies, a total of seven different diagnostic criteria were used. Four studies used radiological criteria for diagnosis (*Alshuft et al., 2016*; *Cottam et al., 2016*; *Cottam et al., 2018*; *Guo et al., 2021*), three used the Clinical Classification of American College of Rheumatology together with the Kellgren-Lawrence classification (*Barroso*
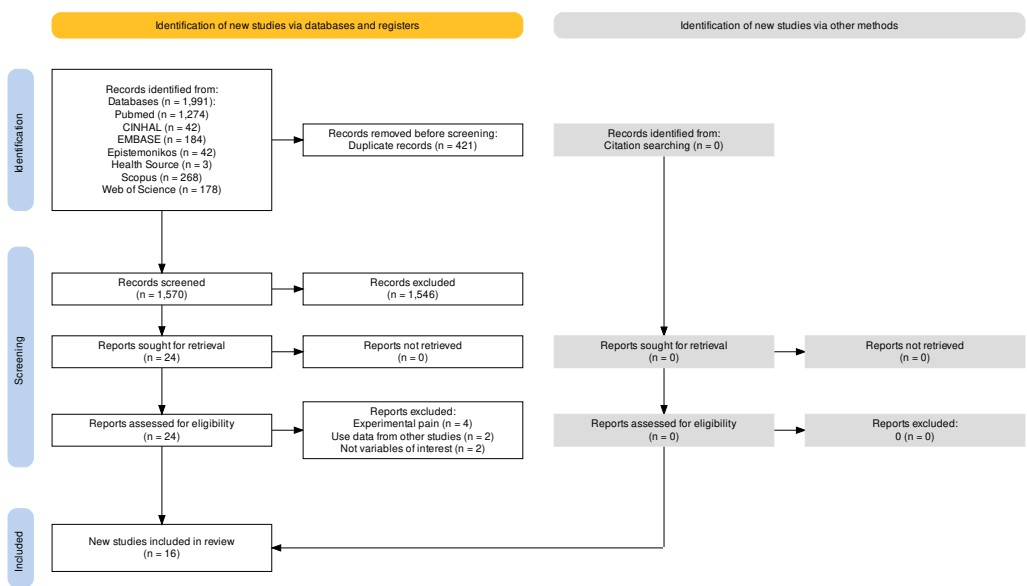

**Figure 1   Flowchart.**

*et al., 2020*; *Barroso et al., 2021*; *Cheng et al., 2022*), two exclusively used the Clinical Classification of the American College of Rheumatology (*Mao et al., 2016*; *Liao et al., 2018*), two exclusively used the Kellgren-Lawrence classification (*Ushio et al., 2020*; *Kang et al., 2021b*), two used medical criteria (*Baliki et al., 2011*; *Baliki et al., 2014*), one applied medical together with radiological criteria (*Lan et al., 2020*), one used the Chinese Guidelines for Diagnosis and Treatment of Osteoarthritis (*Gao et al., 2022*), and one study did not report the diagnostic criteria (*Lewis et al., 2018*).

Of the 16 studies included in this review, twelve used MRI as the imaging technique for exploring brain changes in the participants (*Baliki et al., 2011*; *Baliki et al., 2014*; *Alshuft et al., 2016*; *Mao et al., 2016*; *Liao et al., 2018*; *Cottam et al., 2018*; *Lewis et al., 2018*; *Barroso et al., 2020*; *Barroso et al., 2021*; *Guo et al., 2021*; *Kang et al., 2021b*; *Cheng et al., 2022*), 10 used resting state MRI (*Baliki et al., 2014*; *Cottam et al., 2016*; *Cottam et al., 2018*; *Barroso et al., 2020*; *Barroso et al., 2021*; *Lan et al., 2020*; *Ushio et al., 2020*; *Guo et al., 2021*; *Kang et al., 2021b*; *Gao et al., 2022*), whereas no study used EEG or PET (Table 1).

## Structural brain changes in people with knee OA *versus* healthy controls

Eight studies (*Baliki et al., 2011*; *Alshuft et al., 2016*; *Mao et al., 2016*; *Lewis et al., 2018*; *Liao et al., 2018*; *Barroso et al., 2020*; *Guo et al., 2021*; *Kang et al., 2021b*) reported changes in gray matter volume, reported by MRI, in people with knee OA in comparison to healthy controls. A decrease in volume and thickness of the gray matter in the ínsula (*Baliki et al., 2011*; *Alshuft et al., 2016*; *Guo et al., 2021*), the left precuneus cortex (*Alshuft et al., 2016*), precuneus cortex (*Barroso et al., 2020*), hippocampus (*Baliki et al., 2011*; *Mao et al., 2016*; *Guo et al., 2021*); paracentral lobule, middle anterior cingulate cortex (ACC), visual cortex and inferior temporal cortex (*Baliki et al., 2011*); left middle temporal gyrus and

left inferior temporal gyrus (*Kang et al., 2021b*), left temporal pole(*Barroso et al., 2020*), bilateral orbitofrontal cortex, right lateral prefrontal cortex and postcentral cortex (*Liao et al., 2018*), precentral cortex(*Liao et al., 2018*; *Barroso et al., 2020*), and caudate nucleus (*Mao et al., 2016*) was found in people with knee OA in comparison to healthy subjects. Contrarily, people with knee OA presented an increased volume of the gray matter in the medial frontal gyrus (*Barroso et al., 2020*), bilateral nucleus accumbens, amygdala, and ipsilateral primary somatosensory cortex (*Lewis et al., 2018*) compared to healthy controls. No differences in the gray matter at the right and left ACC, and paracingulate gyri was reported between subjects with knee OA and controls (*Barroso et al., 2020*).

One study reported white matter changes evidenced by an increase in fractional anisotropy and a decrease in axial diffusivity, radial diffusivity, and mean diffusivity at the regions of the corpus callosum, corona radiata, superior longitudinal fasciculus, cingulum, and thalamic radiation in people with knee OA in comparison to healthy controls (*Cheng et al., 2022*).

On the other hand, when grouping studies by diagnostic criteria (radiological, clinical, mixed), it can be seen that those studies using a single diagnostic criterion (*i.e.*, radiological or clinical) showed a tendency to identify a decrease in gray matter volume, while those studies using both criteria for the selection of participants showed a tendency to find both an increase and a decrease in gray matter volume.

## Functional brain changes in people with knee OA *versus* healthy controls

Different studies reported brain functional changes, using rs-fMRI, in several brain areas in people with knee OA in comparison to healthy subjects. In particular there was found a decreased connectivity at the left supramarginal gyrus (SMG) (*Baliki et al., 2014*), right anterior insula associated with posterior cingulate cortex, bilateral parietal areas, and superior frontal gyrus (*Cottam et al., 2018*), within the right temporal pole into the inferior frontal gyrus (*Cottam et al., 2018*); and left middle temporal gyrus to superior frontal gyrus, left middle frontal gyrus, and left medial superior frontal gyrus (*Kang et al., 2021b*). On the other hand, local functional activity presents a decrease in the left cerebellum, left precentral gyrus, right superior occipital gyrus (*Guo et al., 2021*), bilateral angular, precuneus, and medial superior frontal gyrus (*Lan et al., 2020*).

An increase in the connectivity at the right anterior insula within the cuneus (*Cottam et al., 2018*), left anterior insular cortex with the right orbitofrontal cortex and subcallosal area, and the right anterior insulate cortex with the right orbitofrontal cortex, subcallosal area, and the bilateral frontal pole (*Ushio et al., 2020*) was found in people with knee OA compared to healthy controls. Moreover, an increase in the local functional activity of left insula and hippocampus (*Guo et al., 2021*), bilateral amygdaloid nucleus and cerebellum posterior lobe in people with knee OA was also reported (*Lan et al., 2020*).

No functional differences in the posterior cingulate cortex (*Cottam et al., 2018*), periaqueductal gray and raphe nuclei were reported in one study (*Gao et al., 2022*) whereas two studies concluded that there were no brain differences at the functional level between patients and controls (*Cottam et al., 2016*; *Barroso et al., 2021*).

Salazar-Méndez et al. (2023), *PeerJ*, DOI 10.7717/peerj.16003

**Table 1  Study characteristics.**

| Study/Year | Journal | Country | Study design | KOA diagnostic criteria | Sample characteristics | Evaluation tool | Main findings | *p*-value |
|---|---|---|---|---|---|---|---|---|
| Alshuft et al. (2016) | PLoS One | United kingdom | Case-control | Radiological | KOA group (*n* = 40, 52.5% female), age = 66.09 ± 8.47 years; control group (*n* = 30, 56.7% female), age = 62.72 ± 7.44 years | MRI | In KOA compared to control group, a thinner cortex in the right anterior insula and left precuneus cortex (long pain duration) was observed. | both *p* < 0.001 |
| Baliki et al. (2011) | PLoS One | United State of America | Case-control | Medical | KOA group (*n* = 20, 20% female), age = 53.50 ± 7.4; control group (*n* = 46, 56.5% female) age = 38.77 ± 12.5 years | MRI | In KOA compared to control group, decreased GM density in the insula, middle ACC, hippocampus, paracentral lobule, visual cortex, and regions of the inferior temporal cortex was observed. | NI |
| Baliki et al. (2014) | PLoS One | United State of America | Case-control | Medical | KOA group (*n* = 14, 42,9% female), age = 58.29 (42–77) years; control group (*n* = 36, 66,7% female), age = 41.36 (21–70) years | MRI and rs-fMRI | In KOA compared to control group, decreased connectivity (via the average spatial representation of the DMN) of left SMG region was observed. | *p* < 0.001 |
| Barroso et al. (2020) | Pain | Portugal | Case-control | Clinical classification of the American College of Reumatology and Kellgren-Lawrence classification | KOA group (*n* = 91, 79.1% female), age = 65.5 ± 6.5; control group (*n* = 36, 55.5% female), age 59.2 ± 8 years | MRI and rs-fMRI | Total neocortical GM volume and GM volume in the right and left ACC and paracingulate gyri were not significantly different between the KOA and control group; In KOA compared to control group, decreasd volumen in left primary motor cortex (precentral cortex), left temporal pole, and GM volumen in the precuneus cortex was observed; and a increased GM volume in the medial frontal gyrus was observed. | *p* < 0.001 |
| Barroso et al. (2020) | Human Brain Mapping | Portugal | Case-control | Clinical classification of the American College of Reumatology and Kellgren-Lawrence classification | KOA group (*n* = 46, 65.2% female), age = 65.3 ± 7.41 years; control group (*n* = 35, 57.1% female), age = 59.5 ± 7.91 years | MRI and rs-fMRI | Global measures of network topology (e.g., clustering coefficient, global efficiency, betweenness centrality) were no significantly different between KOA and control groups. | NI |
| Cheng et al. (2022) | Frontiers in Neurology | China | Case-control | Clinical classification of the American College of Reumatology-Kellgren-Lawrence classification | KOA group (*n* = 166, 76.8% female), age = 52.87 ± 5.23 years; control group (*n* = 88, 63.3% female), age 53.76 ± 4.82 years | MRI | In KOA compared to control group, increased fractional anisotropy and decreased axial diffusivity, radial diffusivity, and mean diffusivity* in the corpus callosum, corona radiata, longitudinal fasciculus, cingulum, and thalamic radiation were observed. | *p* < 0.05 |
| Cottam et al. (2016) | NeuroImage: Clinical | United kingdom | Case-control | Radiological | KOA group (*n* = 26, 53.8% female), age = 67.0 (45–84) years; control group (*n* = 27, 66.7% female), age 64.5 (43–80) years | rs-fMRI | Global GM cerebral blood flow was not significantly different between KOA and control group. | *p* > 0.05 |

*(continued on next page)*

Salazar-Méndez et al. (2023), *PeerJ*, DOI 10.7717/peerj.16003

**Table 1** (*continued*)

| Study/Year | Journal | Country | Study design | KOA diagnostic criteria | Sample characteristics | Evaluation tool | Main findings | *p*-value |
|---|---|---|---|---|---|---|---|---|
| *Cottam et al. (2018)* | Pain | United kingdom | Case-control | Radiological | KOA group (*n* = 25, 52% female), age = 65.0 (48–84) years; control group (*n* = 19, 57.9% female), age 65.5 (51–80) years | MRI and rs-fMRI | In KOA compared to control group, increased in the right anterior insula functional connectivity within the cuneus and decreased in right anterior insula functional connectivity in areas associated with the DMN including the posterior cingulate cortex, bilateral parietal areas, and the superior frontal gyrus were observed; In KOA compared to control group, reduced CEN functional connectivity in a single cluster that extended superiorly from the right temporal pole into the inferior frontal gyrus was observed; posterior cingulate cortex functional connectivity displayed a similar extent of the DMN between KOA and control group. | *p* < 0.05 |
| *Gao et al. (2022)* | Frontiers in Neurology | China | Pre-experimental | Chinese Guideline for Diagnosis and Treatment of Osteoarthritis (2021 edition) | KOA group (*n* = 15, 53.3% female), age = 59.13 ± 10.27 years; control group (*n* = 15, 73.3% female), age = 58.53 ± 8.15 years | rs-fMRI | Periaqueductal gray and raphe nuclei were not significantly different between KOA and control groups at pre-acupuncture. | NI |
| *Guo et al. (2021)* | Frontiers in Human Neuroscience | China | Case-control | Radiological | KOA group (*n* = 13, 100% female), age = 55.5 ± 5.5 years; control group (*n* = 13, 100% female), age = 53.9 ± 5.6 years | MRI and rs-fMRI | In KOA compared to control group, reduced GM volume in the bilateral insula and bilateral hippocampus was observed; In KOA compared to control group, reduced fractional ALFF in the left cerebellum, left precentral gyrus, and right superior occipital gyrus increased was observed; and increased fractional ALFF in the left insula and bilateral hippocampus was observed. | *p* < 0.001 |
| *Kang et al. (2021b)* | Brain and behavior | China | Case-control | Kellgren-Lawrence classification | KOA group (*n* = 37, 91.9% female), age = 71.6 ± 5.6; control group (*n* = 37, 81.1% female), age = 69.5 ± 5.1 | MRI and rs-fMRI | In KOA compared to control group, decreased GM volume in the left middle TG and left inferior TG was observed; In KOA compared to control group, decreased resting state-functional connectivity in the left middle TG to the superior FG, left middle FG, and left medial superior FG was observed. | *p* < 0.05 |
| *Lan et al. (2020)* | Frontiers in Neurology | China | Pre-experimental | Medical and radiological | KOA group (*n* = 23, 65.2% female), age = 71.2 ± 4.2 years; control group (*n* = 23, 60.9% female), age = 71.4 ± 4.1 years | rs-fMRI | In KOA compared to control group, decreased ALFF in the bilateral angular, precuneus, and medial superior frontal gyrus was observed; In KOA compared to control group, increased ALFF in the bilateral amygdaloid nucleus and cerebellum posterior lobe was observed. | *p* < 0.001 |
| *Lewis et al. (2018)* | Pain Medicine | New Zealand | Pre-experimental | NI | KOA group (*n* = 29, 51.7% female), age = 68.0 ± 10.0 years; control group (*n* = 18, 38.9% female), age = 71.0 ± 8.0 years | MRI | In KOA compared to control group, an increase in the GM volume bilaterally in the nucleus accumbens (NAc) and amygdala, and in the ipsilateral primary somatosensory cortex (S1) was observed. | *p* < 0.01 |

Salazar-Méndez et al. (2023), *PeerJ*, DOI 10.7717/peerj.16003

**Table 1** (*continued*)

| Study/Year | Journal | Country | Study design | KOA diagnostic criteria | Sample characteristics | Evaluation tool | Main findings | *p*-value |
|---|---|---|---|---|---|---|---|---|
| *Liao et al. (2018)* | Medicine | China | Case-control | Clinical classification of the American College of Reumatology | KOA group ($n = 30$, 86.7% female, age = 56.5 ± 6.8 years; control group ($n = 30$, 86.7% female), age = 55.2 ± 5.7 years | MRI | In KOA compared to control group, a decrease in GM volumne in several cortical structures including the bilateral orbital frontal cortex, the right lateral prefrontal cortex, the precentral and part of postcentral cortex was observed. | $p < 0.05$ |
| *Mao et al. (2016)* | Frontiers in Aging Neuroscience | China | Case-control | Clinical classification of the American College of Reumatology | KOA group ($n = 26$, 84.6% female), age = 55.5 ± 9.1; control group ($n = 31$, 83.9% female), age = 53.1 ± 6.4 years | MRI | In KOA compared to control group, smaller volumes of caudate nucleus and hippocampus were observed. | $P = 0.004$ |
| *Ushio et al. (2020)* | Journal of Pain Research | Japan | Case-control | Kellgren-Lawrence classification | KOA group ($n = 19$, 100% female), age = 73.2 ± 5.1 years; control group ($n = 15$, 100% female), age = 74.9 ± 4.6 years | rs-fMRI | In female volunteers with chronic severe KOA compared to control group, the left anterior insular cortex showed stronger resting state-functional connectivity with the right orbitofrontal cortex and the subcallosal area, and the right anterior insulate cortex showed stronger resting state-functional connectivity with the right orbitofrontal cortex, subcallosal area, and the bilateral frontal pole. | $p < 0.005$ |

**Notes.**

KOA, Knee Osteoarthritis; MRI, Magnetic Resonance Imaging; ACC, Anterior Cingulate Cortex; NI, no informed; DMN, Default Mode Network; SMG, Supramarginal Gyrus; rs-fMRI, Rest State-Functional MRI; GM, Gray Matter; CEN, Central Executive Network; ALFF, Amplitude of Low-Frequency Fluctuation; TG, Temporal Gyrus; FG, Frontal Gyrus; NAc, Nucleus Accumbens.

Salazar-Méndez et al. (2023), *PeerJ*, DOI 10.7717/peerj.16003

Peerj

**Table 2  Quality assessment of studies using Newcastle-Ottawa scale for case-control studies.**

| Study ID | Selection | | | | Comparability | | Exposure | | | Total (max = 9 ★) |
|---|---|---|---|---|---|---|---|---|---|---|
| | Case definition (★) | Representativeness of the cases (★) | Selection of Controls (★) | Definition of Controls (★) | (★★) | Ascertainment of exposure (★) | Same method of ascertainment for cases and controls (★) | Non-Response rate (★) | | |
| Alshuft et al. (2016) | ★ | – | – | ★ | – | ★ | ★ | ★ | 5 |
| Baliki et al. (2011) | – | – | – | – | ★★ | ★ | ★ | – | 4 |
| Baliki et al. (2014) | – | – | – | – | – | ★ | ★ | ★ | 3 |
| Barroso et al. (2020) | ★ | – | ★ | ★ | – | ★ | – | ★ | 5 |
| Barroso et al. (2020) | ★ | – | ★ | ★ | ★★ | ★ | – | – | 6 |
| Cheng et al. (2022) | ★ | – | – | ★ | ★★ | ★ | – | – | 5 |
| Cottam et al. (2016) | ★ | – | – | – | – | ★ | ★ | – | 3 |
| Cottam et al. (2018) | ★ | – | – | – | ★★ | ★ | – | – | 5 |
| Gao et al. (2022) | ★ | – | ★ | – | – | ★ | – | ★ | 4 |
| Guo et al. (2021) | ★ | – | – | ★ | ★★ | ★ | – | ★ | 6 |
| Kang et al. (2021b) | ★ | – | – | – | ★★ | ★ | – | ★ | 5 |
| Lan et al. (2020) | ★ | – | ★ | ★ | ★★ | ★ | – | ★ | 7 |
| Lewis et al. (2018) | – | ★ | – | ★ | ★★ | ★ | – | ★ | 6 |
| Liao et al. (2018) | ★ | – | – | ★ | ★★ | ★ | – | ★ | 6 |
| Mao et al. (2016) | ★ | – | ★ | – | – | ★ | – | ★ | 4 |
| Ushio et al. (2020) | ★ | – | – | ★ | ★★ | ★ | – | ★ | 6 |

Finally, grouping the studies by diagnostic criteria (radiological, clinical, mixed) revealed that those studies that applied radiological criteria presented great heterogeneity in their findings since a decrease, increase, combined changes, and no changes in brain functionality were identified, while those studies with clinical selection criteria presented less heterogeneity, identifying either a decrease or no changes in brain function. On the other hand, when using both diagnostic criteria (*i.e.*, radiological and clinical), an increase in functionality was identified.

## DISCUSSION

The objective of this review was to examine the available evidence on the structural and functional brain changes occurring in people with knee OA in comparison with healthy controls. Several structural changes in the gray and white matter and functional changes in brain connectivity and activation were in people with knee OA identified by MRI and functional by rs-fMRI, while no study used EEG or PET for assessment. However, these findings must be interpreted with caution due to the heterogeneity of the diagnostic criteria used in the included studies, for example, studies using a single criterion have a tendency to identify a decrease in gray matter volume, while those studies using both criteria radiological and clinical criteria for participant selection report both an increase and decrease in gray matter volume, which may impact comparability due to differences between the criteria used (*Schiphof et al., 2008*). Furthermore, the direction of these changes varied between studies, but most appear to reflect alterations in the pain matrix in this population. In particular, two studies did not identify any differences between individuals with knee OA and controls (*Cottam et al., 2016*; *Gao et al., 2022*). This may be because one study compared the global cerebral blood flow between the two groups and not the local changes, so could not identify areas in which flow increased and others in which it decreased (*Cottam et al., 2016*), while the other used a small sample, which may not have been sufficient to identify differences between groups (*Gao et al., 2022*).

It has been shown that the presence of chronic pain can induce morphological changes in the brain (*Woolf & Salter, 2000*; *May, 2008*; *May, 2011*; *Farmer, Baliki & Apkarian, 2012*). Furthermore, it is widely accepted that brain regions work in synergy and that their activity can be grouped into several large-scale neural networks (*Fox et al., 2005*; *Mesmoudi et al., 2013*). Therefore, determining what are the changes that occur at the brain level, both in its morphology and in its functioning, in patients with knee OA, allows us to understand the underlying neural mechanisms of the persistence of pain in this condition. The results of this review showed that multiple brain areas can present structural changes in people with knee OA. These findings are in line with similar reviews conducted in other chronic pain populations. In particular, a reduction of gray matter at the cingulate cortex, inferior temporal cortex, hippocampus, nucleus accumbens, amygdala and primary somatosensory ipsilateral ipsilateral was found in people with OA (*Cauda et al., 2014*; *Pedersini et al., 2022*). Similarly, a reduction in gray matter was reported at the somatosensory areas, pre and post central gyrus, hippocampus, insula, and dorsolateral prefrontal cortex in individuals with chronic low back pain (CLBP) (*Cauda et al., 2014*; *Kregel et al., 2015*).

Regarding functional brain changes, our review showed a disparity in behavior between different brain areas. That is, some regions (*e.g.*, right anterior insula within the cuneus, and left anterior insular cortex with the right orbitofrontal cortex) presented a greater connectivity, while others (*e.g.*, SMG, and right anterior insula associated with posterior cingulate cortex) showed the opposite. In people with CLBP, an increased activation of the PFC, amygdala, cingulate cortex, and insula has been reported (*Kregel et al., 2015*) whereas a higher default mode network (DMN) connectivity with the insula has been found in people with fibromyalgia (*Napadow et al., 2010*). On the other hand, a decrease in connectivity between the left insula and the fronto-orbital cortex has been shown in individuals with chronic pain, which worsens the functions of the attention network (*Yoshino et al., 2021*).

Many of the included studies in this review agree that knee OA pain produces structural and functional alterations at the neural (DMN) components (*Baliki et al., 2014*; *Alshuft et al., 2016*; *Lan et al., 2020*; *Barroso et al., 2021*; *Kang et al., 2021b*) such as the precuneus, median temporal gyrus and medial prefrontal cortex (mPFC), as demonstrated in other chronic pain conditions (*e.g.*, CLBP and complex regional pain syndrome (*Baliki et al., 2014*). In addition, several neural networks related to the generation, perception, and regulation of emotions and behavior have been identified. These networks involve areas such as the prefrontal cortex, insula, cingulate gyrus, temporal gyrus, supplementary motor area, amygdala, and periaqueductal gray (*Simons, Elman & Borsook, 2014*; *Morawetz et al., 2020*). In the present study, alterations were identified in these areas, indicating that the neural networks of emotions and behavior are affected in individuals with knee OA.

Furthermore, the large number of affected areas and the behavior of these changes may indicate that knee OA not only generates local changes in functional connectivity, but may also cause a global reorganization of brain networks, demonstrated by the changes obtained from studies including EEG (*Ta Dinh et al., 2019*).

From the aforementioned findings, it is evident that chronic pain can generate both common and specific structural and functional changes in the brain neural networks, dependent on the pathology that affects the person (*Cauda et al., 2014*) and denote the complexity of the neural mechanisms underlying chronic pain (*Baliki et al., 2014*). The results of this review also demonstrated that knee OA pain affects brain areas responsible of the sensory-discriminative, cognitive and affective dimensions of pain (*Hazra et al., 2022*). Concretely, structural and functional alterations in different somatosensory and motor brain regions, such as the precentral and postcentral gyrys, paracentral gyrus (where the primary somatosensory and motor areas from the lower limb are located), cerebellum, and basal ganglia were found and were related to the perception-motor response of pain (*Fenton, Shih & Zolton, 2015*; *Hazra et al., 2022*). Therefore, the frontal cortex also has an important role in the integration of pain sensation, since it is responsible for behaviours related to pain after receiving information from other areas of the brain responsible for processing pain information (*Fenton, Shih & Zolton, 2015*).

Other regions, such as the ACC, thalamus, insular cortex, and amygdala are responsible for the sensory-discriminative components of pain given their specific function and reciprocal connections (*Fenton, Shih & Zolton, 2015*; *Kang et al., 2021a*; *Hazra et al., 2022*;

*Hoskin & Talmi, 2023*). These three regions are also connected to the thalamus, through the paleospinothalamic nociceptive pathway, which is involved in aspects such as attention and mood related to pain (*Horn et al., 2014*).

In addition, the prefrontal and temporal regions, the amygdala, hippocampus, and the basal ganglia are responsible for cognitive domains such as memory, attention, knowledge, and understanding (*Kuner & Kuner, 2021*; *Hazra et al., 2022*). On the other hand, the cingulate cortex, the orbitofrontal cortex, the amygdala, the insular cortex, and the basal ganglia are also involved in the affective aspects of pain perception (*Borsook et al., 2010*; *Hazra et al., 2022*). Furthermore, with respect to the regions involved in the modulatory aspect of the brain matrix of pain (*Hazra et al., 2022*), the current review identified that the prefrontal cortex and cingulate cortex regions are affected in people with knee OA.

The results from the studies included in this review reveal that patients with chronic pain due to knee OA experience alterations in brain structure and function, particularly in areas related to the neuromatrix of pain. These findings suggest that the persistent pain experienced by this population may lead to a distinctive brain signature characterized by structural and/or functional reorganization across multiple areas and connections. Determining precisely the multiple changes in this chronic condition that affects millions of people around the world is relevant since it would allow for more precise objective diagnoses (*Baliki et al., 2011*) and guide the most appropriate treatments for individual needs.

It is important to note that the current review did not yield studies using EEG or PET in patients with knee OA. Specifically, it may be useful to implement EEG together with MRI (*Ta Dinh et al., 2019*) since the former has high temporal resolution (*Levitt & Saab, 2019*), while the latter allows a greater understanding of the spatial structural aspects of the brain (spatial resolution) (*Nunez, Srinivasan & Fields, 2015*; *Lenoir et al., 2020*). Moreover, EEG may represent a brain-based marker of pain given its safety, cost-effectiveness, availability, and potential mobility (*Ta Dinh et al., 2019*) thus allowing to recognize abnormal patterns in brain electrical activity that could be targeted with novel therapeutic strategies (*Accou et al., 2023*) such as non-invasive brain stimulation techniques (*Polanía, Nitsche & Ruff, 2018*). On the other hand, PET studies would enable us to understand the anatomical distribution of physiological processes involved in the perception and modulation of pain (*Dasilva, Zubieta & Dossantos, 2019*). Therefore, this technique could serve as a valuable complement in the study of chronic pain in knee OA, aiding in the determination of the dynamic functioning of the brain systems involved.

## Research implications

Although changes in the structure and function of the brain (neuroplastic changes) have been identified in people with osteoarthritis of the knee, studies have not looked at these changes by subclassifying them by severity, so future research could integrate subgroup analyses according to the severity of the osteoarthritis of the knee. In addition, most of the studies have a moderate methodological quality, so the findings should be taken with caution; therefore, future studies should be more rigorous, especially in representativeness of cases, using the same precision method for cases and controls, and control selection. On

the other hand, EEG has not been included as a tool to assess brain function, as measured by electrical activity, specifically in people with knee osteoarthritis. Therefore, future studies should assess brain electrical behaviour and changes using EEG, as this technique has advantages in its feasibility of use and the data may represent a brain-based marker of pain.

### Strengths and limitations

A strength of this scoping review is that it was performed systematically; each stage was conducted independently by two reviewers, and a broad and sensitive search strategy was implemented to find studies that reported differences in brain structure and functionality between individuals with knee OA and healthy subjects. In addition, this review identifies an important gap in the literature regarding tools to assess brain functionality, since it was identified that EEG has not been used in patients with knee OA, a technique with excellent temporal resolution. Some limitations should be acknowledged. First, only five databases were integrated, so potential studies from other databases may not have been included. Second, subgroups were not made according to the duration of knee arthritis that allow identifying the evolution of the brain changes of the pathology.

## CONCLUSIONS

Our findings indicate that people with knee OA, compared to healthy subjects, present structural differences in specific areas of the brain responsible for comprehensive pain processing, as assessed by MRI. Furthermore, people with knee OA showed changes in functionality (activity and connectivity) of brain areas comprising the pain matrix as evaluated with rs-fMRI. Future research should consider evaluating brain functionality in people with knee OA with EEG due to the economic and safety advantages that it presents.

### Funding
The authors received no funding for this work.

### Competing Interests

Guillermo Méndez-Rebolledo is an Academic Editor for PeerJ. The other authors declare that they have no competing interests.

### Author Contributions

- Joaquín Salazar-Méndez conceived and designed the experiments, performed the experiments, analyzed the data, prepared figures and/or tables, authored or reviewed drafts of the article, and approved the final draft.
- Iván Cuyul-Vásquez conceived and designed the experiments, performed the experiments, authored or reviewed drafts of the article, and approved the final draft.
- Nelson Viscay-Sanhueza conceived and designed the experiments, performed the experiments, authored or reviewed drafts of the article, and approved the final draft.

- Juan Morales-Verdugo conceived and designed the experiments, performed the experiments, analyzed the data, authored or reviewed drafts of the article, and approved the final draft.
- Guillermo Mendez-Rebolledo conceived and designed the experiments, performed the experiments, analyzed the data, prepared figures and/or tables, authored or reviewed drafts of the article, and approved the final draft.
- Felipe Ponce-Fuentes analyzed the data, authored or reviewed drafts of the article, and approved the final draft.
- Enrique Lluch-Girbés analyzed the data, authored or reviewed drafts of the article, and approved the final draft.

## Data Deposition

This is a literature review.

## Supplemental Information

Supplemental information for this article can be found online at http://dx.doi.org/10.7717/peerj.16003#supplemental-information.

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
