# Peer review of "Structural and functional brain changes in people with knee osteoarthritis: a scoping review"

_PeerJ, doi:10.7717/peerj.16003_

## Round 0.1 · original submission · Minor Revisions

There are minor concerns about the scoping review that need to be looked at and revised.

**Language Note:** The review process has identified that the English language must be improved. PeerJ can provide language editing services - please contact us at copyediting@peerj.com for pricing (be sure to provide your manuscript number and title). Alternatively, you should make your own arrangements to improve the language quality and provide details in your response letter. – PeerJ Staff

Reviewer 1 ·

Basic reporting

The authors provided a clear and comprehensive description of the study design, outlined the screening process and inclusion criteria for previous studies.

Experimental design

In the study design section, the author has provided the overall number of participants across all studies, it would be more informative to include the number of participants in each individual study. This level of detail is essential for readers to gauge the sample size and understand the robustness of the findings from each study included in the review. It will also be helpful to provide a summary table to list the findings of each study on structural brain change and functional brain change.

Validity of the findings

Within the study characteristics section, it is worth noting that out of the sixteen studies included, there are a total of seven distinct diagnosis criteria utilized. This raises an important question regarding the comparability of patients across these different studies, considering the variability in diagnostic criteria. It would be beneficial to address how the researchers addressed this potential source of heterogeneity and ensured a reasonable level of comparability among patients across the diverse set of studies. By discussing the strategies employed to address this issue, readers will gain a better understanding of the potential impact of varying diagnostic criteria on the study outcomes and the overall validity of the findings.

In identifying functional brain changes in people with knee OA versus healthy controls, While some studies suggest a difference between patients and controls, two studies indicate no difference. It will be helpful to address this inconsistency in the review and discuss potential reasons for the divergent results.

·

Basic reporting

This review article was written with clarity and systematic. Figures and tables were scientifically presented. References are balance from current and past studies.

This review focuses on structural and functional neuroscience.

Within PeerJ scope.

In my knowledge, pain perception study using neuroimaging modalities are of current interest. There is no similar review article published. Topic covered in this review is interesting and new, suggested for its acceptability.

Comment on Language: The introduction part is clear. However, the English use should be improved to ensure its comprehension easier.

Experimental design

Method of searching: Rigorous investigation was performed to a high technical & ethical standard. Authors used a robust method in searching articles, as well as assessing the quality of studies. They covered a broad resources. Utilising NOS to measure the quality of studies ensures the process to be done systematically and that the information was presented with sufficient detail.

Screening and study eligibility criteria was performed by systematic review manager (www.rayyan.ai) assists authors to perform their reviews in a quick and easy manner. Studies written in English and Spanish were considered in this review.

Data extraction was performed systematically by using a standardised form by two reviewers.

Methods were described with sufficient detail and information to replicate.

Survey method was covered fairly with a focus going towards MRI and rs-fMRI. However, keywords related to EEG and PET was missing (such as Phase Locking Value or phase synchronisation which can lead to the information about brain connectivity).

The review is organised coherently.

Validity of the findings

Results were explained concisely. Result of structural brain changes was thoroughly reported.

Functional neuroimaging technique involves several other modalities, like PET and EEG. Considering this, I think authors should mention their scope of study with certain modalities in the mind. If authors had conducted some literature studies about PET and EEG for but didn't find any suitable studies reporting that, it should be mentioned clearly in the report section. The report can shed light to researchers whose their focus falls into EEG or PET, for example.

The absenteeism of studies using EEG for structural changes measurement was mentioned in discussion section. Authors are advised to interpret some of the structural changes effects to their functionalities from cognitive, emotion and behavioural aspects. For example, changes of the attention network due to pain they experienced.

Additional comments

Chronic pain will leave its 'fingerprint' in the structural and electrical brain. Knee OA is one of the common chronic disease inflicts in chronic pain in millions of people worldwide. Authors are suggested to explain about this briefly.

---

## Round 0.2 · accepted · Accept

Happy to note that the manuscript is accepted.

Reviewer 1 ·

Basic reporting

I think that the authors have adequately addressed the comments made by the reviewers in the revised version of the manuscript. Therefore, I have no further comments.

Experimental design

I think that the authors have adequately addressed the comments made by the reviewers in the revised version of the manuscript. Therefore, I have no further comments.

Validity of the findings

I think that the authors have adequately addressed the comments made by the reviewers in the revised version of the manuscript. Therefore, I have no further comments.

·

Basic reporting

This article is now a much improved version, and good to go. Well done. Recommend for publication.

Experimental design

Excellent

Validity of the findings

Excellent

Additional comments

This article is now a much improved version, and good to go. Well done.